# An Examination of the Determination of Medical Capacity under a National Health Insurance Program

**DOI:** 10.3390/ijerph16071206

**Published:** 2019-04-03

**Authors:** Yi-Tui Chen

**Affiliations:** Department of Health Care Management, National Taipei University of Nursing and Health Sciences, 89, Nei-Chiang St. Wan-Hua Dist. Taipei 108, Taiwan; yitui@ntunhs.edu.tw; Tel.: +886-2-238-85-111

**Keywords:** medical capacity, constant elasticity of substitution, National Health Insurance program, medical expenditure, occupancy rate

## Abstract

This paper examines the capacity determination factors of medical services at a national level through the analysis of a mathematical model that maximizes social welfare, which consists of the consumption of private goods and the medical capacity provided by the society. A sensitivity analysis is conducted to investigate the impact of these factors on the medical capacity provided. Furthermore, a case example based on the data provided by the government is presented to discuss the results derived from the theoretical analysis. The results of the sensitivity analysis indicate that individual disposable income, the medical expenditure for each treatment, the level of premium payments, and substitution parameters have a positive impact on medical capacity, while the medical costs and preference parameter negatively affect medical capacity. The results of the correlation analysis based on the data of the case example are consistent with the findings of the theoretical analysis.

## 1. Introduction

The mortality rate has increased steadily in the past several decades in Taiwan, according to statistics released by the Ministry of Health and Welfare [1]. For example, 172,418 deaths occurred in Taiwan in 2016, which marked a 5.4% increase from the number of deaths in the previous year. Furthermore, the number of aging adults (≥65 years old) reached 314,000 by the end of 2017, and the proportion of elderly population is expected to increase from 7% in 1993 to 14% in 2018 [2]. In contrast, the crude birth rate decreased continuously from 9.86% in 2012 to 8.86% in 2016. The decreasing birth rates, greater longevity, higher population of elderly adults and increased environmental pollution, including air and water pollution, may increase the demand for medical services and the utilization of expensive medical resources [3].

Numerous empirical studies have shown a positive relationship between medical capacity and survival for many therapies. Higher capacities of treatment facilities may yield improved survival rates of cancer patients [4,5,6,7]. A study by Joshi et al. [8] found that facility capacity is highly associated with improvements in survival for patients with metastatic renal cell carcinoma (mRCC). Increased capacity requires greater investment in the provision of medical facilities, and consequently increases the cost of medical services. Eventually, the overall regulated premium will rise, and may bring about political objections.

Some previous studies have focused on the location of medical facilities through a variety of research methods. Numerous studies have focused on the location modeling of medical facilities by using operations- and research-based techniques [9,10,11,12,13]. Most of these previous studies indicate more interest in the optimal location of medical facilities based on transportation accessibility, adaptability, and availability, with the objective of maximizing the coverage area of the citizens or reducing the transportation time.

The other stream of previous studies has focused on forecasting the random demand for medical services. The determination of medical capacity may be planned and managed effectively based on an efficient forecasting method. Many researchers have developed methods to forecast the demand for call centers operating in different sectors (e.g., for emergency medical services [14,15,16], for telecommunication [17,18], and for banks) [19,20].

The impact factors affecting the optimal capacity for the provision of medical services may provide valuable information for policymakers. However, many scholars have focused on the micro-level to analyze the medical capacity planning problem. For example, Sariyer [14] employed the newsvendor model in planning the capacities of emergency call centers. Jiang and Seidmann [21] focused on the design of efficient managerial contracts and capacity planning for medical facilities, and analyzed the impact of the linear contracting structure on capacity decisions, service levels, service volumes, and the allocations of costs. Yi et al. [22] developed a generic simulation model to obtain steady-state hospital capacities. In order to reduce the prevalence of preventable childhood diseases in Nigeria, Gai et al. [23] presented a mathematical model to determine hub locations by optimizing a shortest path, and to obtain the optimal capacity of vaccine at the hubs. Some studies have designed a questionnaire to consult with experts by using the Delphi method. For example, Alberti et al. [24] aimed to optimize the prescription center capacity by conducting a Delphi method that aimed at increasing access to therapy for patients with chronic hepatitis C virus (HCV) infection. 

This paper attempts to determine the optimal medical capacity based on a national level through a systematic perspective. A mathematical model is presented to investigate the impact factors by integrating the demand, the supply, and the insurance program. The problem is formulated by maximizing social welfare, which consists of the consumption of private goods and the medical capacity provided under the constraint of a given income. A sensitivity analysis is conducted, and a case example is presented based on real data provided by the governmental statistics bank to demonstrate how the model is applied in practice.

## 2. The Model

This paper assumes that the determination of medical capacity is made by policymakers to maximize social welfare, which consists of the consumption of private goods and the provision of medical services. Citizens with average values for the relevant factors are used to represent their societies. Citizens consume private goods Q each year, and the provision of medical capacity per capita g is available for them when they are sick. The social welfare is measured by a utility function that defines consumer’s preferences for goods or services. A utility function is generally classified into Perfect Substitutes, Perfect Complements, and Cobb–Douglas utility function. Solow [25] presented the constant elasticity of substitution (CES) function for the application in economics fields to model production functions or ordinary consumer choice problems. Later, Arrow et al. [26] applied the CES utility function to consumer theories to describe the preference characterized by a constant elasticity of substitution between two differentiated goods or services. The CES function has a relative advantage, as it covers a much broader spectrum of substitutability between x and y. For ρ = −1, the function collapses to linear functions, representing perfect substitutes. For ρ → ∞, the curve of the CES functions is L-shaped, representing perfect complements. When ρ approaches 0, the CES function converges to the Cobb–Douglas function [27]. As the CES utility function has been widely accepted in research fields, this paper adopted it to serve as the measurement of social welfare. Thus, a constant elasticity of substitution between the provision of medical capacity g and the consumption of private goods Q is employed to describe the welfare of an individual, shown as follows:(1)$$U={\left[\beta Q^{-\rho }+\left(1-\beta \right)g^{-\rho }\right]}^{-\frac{1}{\rho }}$$
where β is a preference parameter between the consumption of private and medical services and ρ is the corresponding substitution parameter with the property ρ>−1.

This paper assumes that the insurance plan covers n people with the probability q of medical services for each person, where *q* represents the disease occurrence rate. The enrollee must pay the health insurance fee for medical services, and the premium payment is assumed to be a proportional rate α of his or her income *y*. When a medical service is completed, a patient must pay a copayment c, and thus, the total expense for the representative citizen is expressed on the right-hand side of Equation (2) and should be equivalent to his or her income.
*y* = *pQ* + *αy* + *qc*(2)
where *p* is the consumer price index. Equation (2) is used as the constraint for Equation (1) to formulate a maximization problem.

In contrast, the revenue R for medical service providers in society is as follows:(3)R=nqm+nqc
where *m* is the medical expenditure for each treatment paid by the National Health Insurance (NHI) program. This paper assumes that the medical cost for each unit of medical capacity h is constant. The total cost of medical services includes the following: labor costs for physicians, nurses, and other technicians; depreciation costs for the facility and devices; administration costs; and utility costs, which are represented by the product of the unit capacity cost h and the total medical capacity G, expressed on the right-hand side of Equation (4). As medical service providers serve as nonprofit organizations, the revenue for medical services should be equal to total costs, shown as follows:
*R* = *nqm* + *nqc* = *hG*(4)
Rearranging Equation (4) yields the following:(5)$$g=\frac{qm+qc}{h}$$
where $g=\frac{G}{n}$. Substituting Equation (5) into Equation (2) yields the following:(6)$$y=\frac{pQ+hg-gm}{1-\alpha }$$
Thus, the problem is formulated and expressed below:(P.1)$$\begin{array}{c} Max\\ Q,g \end{array}{\left[\beta Q^{-\rho }+\left(1-\beta \right)g^{-\rho }\right]}^{-\frac{1}{\rho }}$$
$$\mathrm{Subject}\;\mathrm{to}\;y=\frac{pQ+hg-qm}{1-\alpha }$$
To solve the problem, a Lagrange function is obtained, shown as follows:(7)$$L\;=\;[\beta Q^{-\rho }+(1-\beta )g^{-\rho }{]}^{-\frac{1}{\rho }}\;+\;\lambda \left(y-\frac{pQ+hg-qm}{1-\alpha }\right)$$

Taking the derivatives of Equation (7) with respect to Q and g yields the following:(8)$$-\frac{1}{\rho }{\left[\beta Q^{-\rho }+\left(1-\beta \right)g^{-\rho }\right]}^{-\frac{1}{\rho }-1}\beta \left(-\rho \right)Q^{-\rho -1}-\lambda \frac{p}{1-\alpha }\;=\;0$$
(9)$$-\frac{1}{\rho }{\left[\beta Q^{-\rho }+\left(1-\beta \right)g^{-\rho }\right]}^{-\frac{1}{\rho }-1}\left(1-\beta \right)\left(-\rho \right)g^{-\rho -1}-\lambda \frac{h}{1-\alpha }\;=\;0$$
Solving Equations (8), (9), and (6) yields the optimal solution $g^{*}$, expressed below:(10)$$g^{*}=\left[\frac{\left(1-\alpha \right)y+qm}{P}\right]{\left[{\left(\frac{\beta h}{\left(1-\beta \right)P}\right)}^{\frac{1}{\rho +1}}+\frac{h}{p}\right]}^{-1}$$

Equation (10) indicates that the optimal medical capacity per capita is dependent on the disposable income y, the disease occurrence rate q, the medical expenditure per case *m*, the price index *P*, the cost per unit of medical capacity *h*, and the model’s parameters. By substitution, the analogous optimal solution for $Q^{*}$ could be found. As the aim of this paper focuses on the relevant factors affecting medical capacity, the optimal solution for $Q^{*}$ is deleted. The definitions of symbols are given in Table 1.

In order to examine the influence direction of these factors on the determination of medical capacity, a sensitivity analysis was conducted. By taking derivatives of $g^{*}$ in Equation (10) with respect to the corresponding impact factors, the impact direction was found and is listed in Table 2.

Table 2 demonstrates that the disposable income, the disease occurrence rate, the medical expenditure, the premium rate, and the substitution rate may have a positive impact on medical capacity. In contrast, a higher preference for the consumption of private goods may lead to lower medical capacity. The impact direction of price indexes on medical capacity is uncertain. To provide an in-depth understanding of the relationship between these impact factors and medical capacity, a real case example is presented in Section 4, and an empirical analysis is conducted.

## 3. Research Methods

Taiwan’s National Health Insurance (NHI) program started on 1 March 1995, providing comprehensive coverage, including outpatient care, inpatient care, prescription drugs, dental care, eye care, mental illness treatment, traditional Chinese medicine, and in-home care. Currently, the population in Taiwan is 23,978,460, and the population registered by NHI is 23,832,550 [28]. This implies that insured rate has reached 99.6%. The goal of the NHI program is to control increasing medical costs, to reduce unnecessary length of hospital stays, to eventually enhance medical efficiency, and to avoid wasting resources under a single-payment, universal coverage insurance system [29]. Such a health insurance system is compulsory. However, the copayment rate for each disease is believed to be low and affordable for most citizens. Currently, the overall enrollment in the NHI program is very high, with up to 99.70% of citizens enrolled [30].

The government heavily subsidizes the poor, the ill, and workers in the informal sector to provide a basic standard of health care. Premium exemption is offered to low-income or no-income groups. Currently, the premium for health insurance is dependent on an individual’s income and is borne by the worker (the beneficiary), the employer, and the government. In general, premium payments for formal employees are automatically deducted from their income. The insurance rate in 2016 was 4.69%. In 2016, workers paid 34.95% of the NHI’s total premium revenue, employers 29.05%, and the government 36% [30].

More than 92.8% of health care providers, including hospitals and clinics, contracted with NHI for the supply of medical services at the end of 2016 [31]. To ensure the quality of medical care in Taiwan, the capacity for medical services was assessed to satisfy the health needs in each region of Taiwan. By the end of 2016, a total of 493 hospitals, 21,894 clinics, 7907 pharmacies, and 1318 nursing institutions and other types of institutions were providing medical services [28].

As physicians and nurses play the primary role in providing medical services, in addition to hospital beds, this paper considers (1) the number of beds, (2) the total number of individuals in the workforce, (3) the number of physicians, and (4) the number of nurses to represent the capacity provided in practice. By the end of 2017, medical care institutions that offered general beds, special beds, specially designated beds, and beds in clinics provided a total of 164,590 beds. Figure 1 indicates the historical trend of the number of workforce employees, physicians, nurses, and beds per 10,000 people during 1998–2017. 

In 2017, 10,000 people were served by 69.83 beds, marking an increase from 56.80 beds in 1998 [28]. By the end of 2017, 251,041 individuals in the workforce were practicing in the health profession, including 46,311 physicians, 6685 traditional Chinese medicine doctors, 14,379 dentists, 14,695 pharmacists, and 122,063 nurses and other professionals [32]. This means that the total workforce per 10,000 population increased from 55.41 in 1998, including 12.34 physicians and 16.31 nurses, to 106.50 in 2017, including 19.65 physicians and 51.78 nurses. Figure 1 demonstrates that the growth rate of beds was much lower than that of personnel. The growth rate for the provision of beds per capita from 1998–2017 was 22.94%, which ranked lowest. In contrast, the number of nurses increased 217% for the same period, which ranked highest. The high growth rate of nurses compared to physicians and beds may be attributed to the expanding scopes of practice for nurses [33,34]. The rapid development of medical technology also increased nurses’ workload. Considering the increase in the utilization efficiency of human resources, hospitals prefer the scopes of practice to be overlapping and flexible [35]. Some administrative tasks have been shifted to nurses to reduce health care costs [36]. Thus, the growth rate of nurses is much higher than that of physicians.

The data for the relevant impact factors affecting the determination of medical capacity, which have been provided by governmental statistics, are shown in Table 3. Economic development may be a driving force in the demand for health care. The GDP experienced a higher growth rate of 7–10% in the past several decades. However, the annual economic growth rate decreased to 2–4% in 2012–2016. Table 3 indicates that the disposable income per capita increased from NT$231,611 (US$7471) in 1998 to NT$331,903 (US$10,706) in 2017 [37]. The price has been quite stable in the past several decades, and the price index increased from 84.26 in 1998 to 100.62 in 2017. The growth rate of the consumptive price index fluctuated between −0.87% and 3.52% during the 1998–2017 period [37].

Generally, medical expenses vary greatly for different types of diseases. In general, the expense of inpatient care is higher than that of outpatient care. The expense for cancer-related treatment is very high, approximately three times the average medical expense per person, according to Taiwan’s NHI. Based on the data released by the MOHW [32], per capita health expenditures increased from NT$22,425 in 1998 to NT$46,219 in 2016, with an average growth rate of 2.57% annually. The medical expenditure m listed in Table 3, including inpatient and outpatient care, is based on each case. The medical expenditure per case increased from NT$698 in 2002 to NT$1093 in 2016 for outpatient care, and from NT$37,782 in 2002 to NT$52,391 in 2016 for inpatient care.

The disease occurrence rate q is defined as the average frequency of hospital or clinic visits in a year for each person. Thus, the disease occurrence rate q is calculated by the following:(11)$$q=\frac{F}{n}$$
where *F* is the number of visits to hospitals and clinics. Table 3 demonstrates that the disease occurrence rate for outpatient care increased from 14.12 times in 2002 to 15.36 times in 2016, and for inpatient care from 0.131 cases to 0.141 cases. This finding indicates that, in 2016, each Taiwanese patient visited hospitals or clinics, on average, more than 15 times to receive medical services, and used inpatient medical services approximately 141 times for each 1000 people.

The historical data of medical costs for each unit of capacity are not currently available. As medical equipment and devices are among the major medical costs, this paper employs the sale price index of medical equipment and supplies to represent the medical costs for each unit of medical capacity. The sale price index of medical equipment and supplies decreased from 108.41 in 1998 to 91.61 in 2005, and then fluctuated between 1996 and 2017. The sale price index of medical equipment and supplies in 2017 was 95.24.

In fact, the relevant data released by the government does not fully reveal the limitation of the data calculation. For example, the occurrence rate of diseases is calculated based on the visits per beneficiary issued by NHI. Cases of home nursing care, psychiatric rehabilitation, medical examination referrals commissioned by medical institutions, and repeated prescriptions for patients with chronic illnesses are excluded from the occurrence rate of outpatient care. Inappropriate surveying or incorrect collection technique may lead to bias. Fortunately, the data involving medical expenditure, the occurrence rate of inpatient and outpatient treatment, and total cost is collected by the government census through the insurance program system. This means that sampling error can be avoided. Furthermore, the insured rate has reached 99.6%, implying that the data collected may fully reflect reality.

## 4. Results

Based on the medical capacity data shown in Figure 1 and the impact factors shown in Table 3, a correlation analysis was conducted. As the parameters of preference and elasticity are unknown, the impact of the two parameters on the medical capacity provided is omitted in the correlation analysis in this paper. Furthermore, the premium rate is given and was almost constant over time, and thus the empirical analysis on the impact of the premium rate has also been omitted in the correlation analysis.

The result of the correlation analysis is displayed in Table 4. The correlation coefficient between per capita disposable income and per capita number of beds, per capita number of workforce employees, per capita number of physicians, and per capita number of nurses was 0.8434, 0.9534, 0.9619, and 0.9523, respectively. A strong link was revealed between disposable income and the medical capacity provided. Higher disposable income may expand the demand for medical services, and thus further increase the provision of per capita medical capacity. Several studies have found that wealthy tend to utilize more health services than the poor under a health insurance system [38,39,40]. Wealthier people may care more about health status, and tend to have routine physical examinations. Laaksonen et al. [41] found that participants in the highest income quartile are more likely to participate in occupational health checkups than those in the lowest quartile. In practice, physical examinations have become a popular activity in most developed countries, as evidenced by the health check programs implemented in countries such as the USA, the UK, Holland, and Australia [42]. Thus, an increase in income may drive people to utilize more medical services [43]. The strong relationship presented in this paper between disposable income and medical capacity obtained implies that increased income may increase the demand for medical services, and eventually lead to increased medical capacity.

The price index was also found to be highly associated with the medical capacity provided, with correlation coefficients higher than 0.9 for all four kinds of medical capacity, as shown in Table 4.

The expenditures for both outpatient and inpatient care were also highly associated with the medical capacity provided. All the correlation coefficients were higher than 0.8 for both outpatient care and inpatient care. The correlation coefficient was higher than 0.6 for both outpatient care and inpatient care, as shown in Table 4. This finding implies that hospitals (the service providers) may be encouraged to expand their medical capacity by medical revenue (equivalent to the medical expenditures paid by the NHI program). However, the correlation coefficient for inpatient care was higher. The stronger relationship between hospital beds and inpatient care than between hospital beds and outpatient care may be explained by the fact that hospital beds are utilized in inpatient care only.

The sale price index of medical equipment and supplies as a proxy for total costs was found to have a negative relationship with per capita medical capacity, with correlation coefficients ranging between −0.3170 and −0.4578. This finding implies that the cost of medical equipment and supplies may encourage hospitals to invest in more facilities. The empirical results shown in Table 4 fully coincide with the theoretical analysis shown in Table 2.

## 5. Discussion

This paper considers a citizen with mean values for demographic characteristics to represent the society. This implies that the distribution of the demographic characteristics is normal. If the gap between mean and median value increases, the optimal capacity needs to be adjusted. For example, the mean and median value of Taiwan’s personal disposable income in 2016 were NT$323,490 and NT$310,807, respectively. Comparing the optimal capacity derived by median value and mean value, the following equation derived from Equation (10) is obtained.
(12)$$\frac{g^{\#}}{g^{*}}=\frac{\left(1-\alpha \right)y^{\#}+qm}{\left(1-\alpha \right)y^{*}+qm}$$
where the $y^{\#}$ and $y^{*}$ are the mean and median disposable income, and $g^{\#}$ and $g^{*}$ are the optimal capacity obtained by the median and the mean value, respectively. Applying the data of disposable income, disease occurrence rates, and medical expenditure shown in Table 3 to Equation (12) gives a result of $\frac{g^{\#}}{g^{*}}=$ 96.36%. Such a result implies that the optimal capacity may be reduced by 3.67% if calculated by median values compared to mean values. Basically, high socioeconomic inequalities may have a negative impact on the maintenance of a health care system. As the gap between the mean and the disposable income is only 3.9%, economic inequality seems not so serious in Taiwan; thus, the optimal value of medical capacity based on a mean value may be acceptable.

Given the financial risk, the NHI administrator attempts to reduce per capita medical expenditures if premium payments cannot be increased. A cost-control mechanism should be required to regulate physicians’ prescriptions of unnecessary quantities and types of pharmaceuticals, to reduce the risk of patient overuse or misuse. Based on the derived results, this paper suggests that the effort to improve the efficiency of medical services might focus on both sides of medical services, i.e., the demand side and the supply side. On the demand side, the reduction in per capita medical expenditures may be the primary strategy. The policy implemented by the government may affect changes in medical demand. For example, Taiwan implemented a screening program for breast cancer, oral cancer, colorectal cancer, and cervical cancer by providing subsidies. Each citizen aged 40–65 years is granted permission to engage in such an examination every three years, and those aged older than 65 years can visit hospitals for an annual routine checkup provided by the government. The national cancer screening program started providing Pap smears for women in 1985, screenings for breast cancer and oral cancer in 1999, and for colorectal cancer in 2004 [44].

Such a program expands the consumption of medical services, and leads to an increase in per capita medical expenditures. In 1991, the per capita medical expenditure was only NT$10,555. It continued to grow, and reached NT$46,219 in 2016, as shown in Figure 2. A positive relationship between per capita medical expenditures and disposable income was found. This implies that economic development may push up the demand for medical services. The stable growth of the per capita expenditure caused by these preventive services implemented by the government may encourage hospitals to invest in highly advanced facilities, and lead to a continuous increase in medical capacity. 

Furthermore, a high disease occurrence rate may result in an increase in the consumption of medical services, and thus increase the investment in medical equipment and the employment of relevant personnel. The weaker relationship between outpatient care and medical capacity than between inpatient care and medical capacity demonstrates that the utilization of personnel for outpatient care is more flexible. This finding indicates that the demand for outpatient care is more flexible, and may be reduced through appropriate measures. Thus, preventive programs probably need to be reconsidered with respect to their appropriateness, or substituted by disease awareness and education programs.

On the hospital side, overall rising medical costs are caused by the low occupancy rate of medical facilities and wasted medical services. The occupancy rate of hospital beds for both general beds and special beds slightly increased during the period of 1998–2017. The occupancy rate of general beds increased from 62.68% in 1997 to 71.99% in 2017, and that of special beds grew from 47.49% in 1998 to 66.68% in 2017 [32]. Some studies suggest an absolute maximum bed occupancy rate of between 82% and 85% to reduce infection in hospitals [45]. A literature review conducted by Tierney and Conroy [46] based on studies published between 1997 and 2012 found that optimal occupancy rate for intensive care units is approximately 70–75%. In practice, economic growth may stimulate the public to be more concerned on health status and, as a consequence, more outpatient care and body examinations may be demanded. Outpatient care may detect the existence of major diseases at earlier stages and avoid the demand for beds. However, an aging society is characterized by high demand for the treatment of chronic diseases and increasing demand for beds.

Some studies find that overcrowding and understaffing may reduce patient safety and increase the burden of healthcare workers. An empirical study by Medley et al. [47] confirmed a significant relationship between crowding and violence toward staff through a multivariate logistic regression analysis. Sprivulis et al. [48] and Richardson [49] found a positive association between crowding and mortality. Tierney and Conroy [46] argue that a uniform target occupancy rate for all intensive care units is problematic, due to a range of influencing factors. However, some researchers still suggest that the occupancy rate is an appropriate index to measure crowding in emergency departments [47,50]. This paper suggests that an appropriate threshold value is still required to evaluate the operating efficiency of hospital beds, based on the objective characteristics of each bed. The data released by MOHW [32], in association with the occupancy of hospital beds in 2017, are listed in Table 5, in which the occupancy rate of several types of beds falls below 80%. In particular, the occupancy rate of beds for the treatment of general chronic disease is still very low. For example, the occupancy rate of chronic T.B. beds is only 20.14%, and that of chronic general beds is 54.48%. Medical service providers may evaluate the impact of the occupancy rate on cost and patient safety. Thus, a flexible and effective tool to improve management and resource utilization may be required.

## 6. Conclusions

This paper explores a theoretical model for the determination of optimal capacity, and describes the results of a sensitivity analysis. By using the empirical data provided by the government, the correlation analysis confirms the feasibility of applying such a model in the practical world to support health management, in association with investment in hospital beds or the employment of personnel. The theoretical model presented in this paper may be extended to the level of different medical departments by applying the data of impact factors for different institutions. This approach enables the determination of optimal capacity through a mathematical model by using practical data in a practical world. Such an approach considers both theoretical and practical methods.

Although some studies have attempted to solve the solution for the capacity planning problem, they completely focused on a micro-level [14,21,22,23,51]. This paper attempts to determine the optimal medical capacity on a state level rather than a hospital level, based on a systematic perspective. The model presented in this paper integrates the perspectives of the policy maker, the supplier (medical service providers), the demanders (the patients), and the health-care insurance systems. Thus, it provides more valuable information for policy makers to make adjustments to medical care policy.

The effective management of healthcare resources through the optimization of medical capacity is a vital way to control costs. The planning and management of health insurance systems are important issues for policymakers. To provide sufficient medical services, providers should carefully examine the required capacity of medical facilities, including the number of physicians, nurses, and other staff, beds, and other equipment. This paper finds that several factors and some uncertain parameters play important roles in affecting the determination of optimal capacity for the provision of medical services.

From a theoretical point of view, the policymaker needs to examine the factors affecting the capacity provided. The model presented in this paper may be employed by policymakers in the determination of medical capacity through the planning of premium rates, medical expenditures, and disease occurrence rates, and the consideration of disposable incomes and price indexes. In the meantime, the policymaker also needs to estimate/forecast possible changes in demographic characteristics in society, as growing longevity may increase the demand for medical services. Considering the increased medical expenditures due to longevity and other social factors, future research may be extended to an analysis of the association between the impact of demographic characteristics and the demand for medical services.

## Figures and Tables

**Figure 1 ijerph-16-01206-f001:**
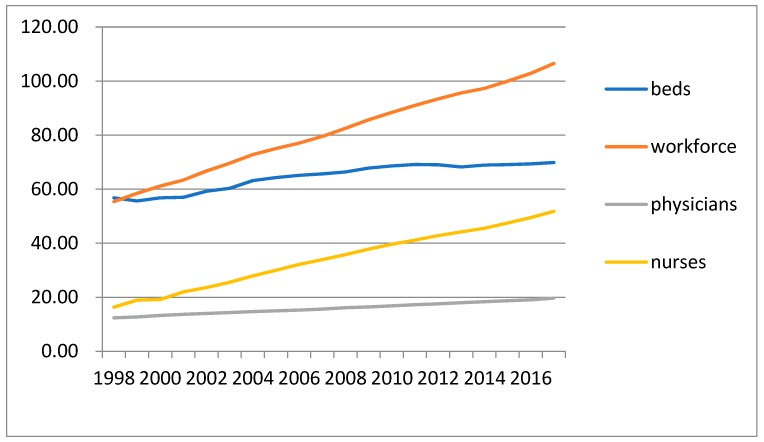
The trend of beds, workforce, physicians, and nurses provided per 10,000 population in Taiwan.

**Figure 2 ijerph-16-01206-f002:**
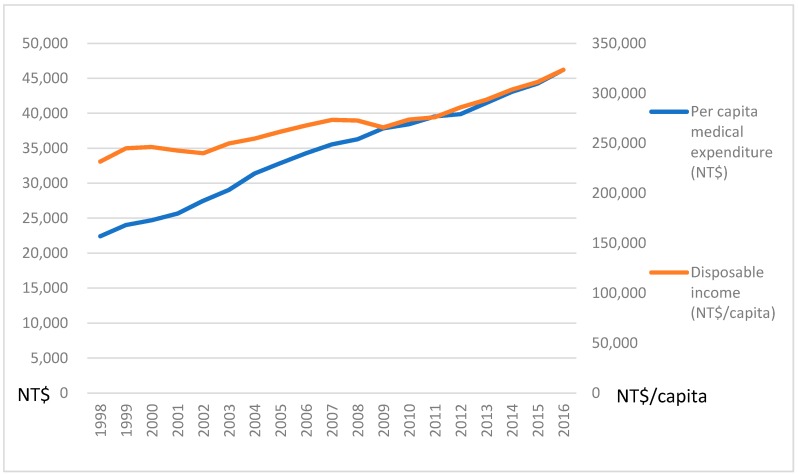
The trend of per capita medical expenditure and disposable income.

**Table 1 ijerph-16-01206-t001:** The list of symbols.

Symbols	Definition
β	A parameter between the consumption of private and medical services
Q	Private goods
ρ	Substitution parameter
*g*	The provision of medical capacity
*y*	Disposable income
p	The consumer price index
α	The proportion of
q	The disease occurrence rate
c	Copayment paid by patient
n	e amount of people covered by insurance plan
R	The revenue of medical service
m	The medical expenditure for each treatment paid by the National Health Insurance

**Table 2 ijerph-16-01206-t002:** The directions of the factors affecting medical capacity.

Symbols	*y*	*q*	*M*	*P*	*h*	α	β	ρ
$g^{*}$	+	+	+	?	−	+	−	+

Remarks: * denotes the optimal solution.

**Table 3 ijerph-16-01206-t003:** The data for disposable income, price index, medical expenditure, disease occurrence rate, and the total cost for each bed.

Title	Disposable Income *y*	Price Index *p*	Expenditure *m*		Occurrence Rate *q*		Total Costs *h*
			Inpatient	Outpatient	Inpatient	Outpatient	
1998	231,611	84.26	n.a.	n.a.	n.a.	n.a.	108.41
1999	244,918	84.41	n.a.	n.a.	n.a.	n.a.	101.05
2000	246,256	85.47	n.a.	n.a.	n.a.	n.a.	95.04
2001	242,640	85.46	n.a.	n.a.	n.a.	n.a.	101.56
2002	239,978	85.29	698	37,782	14.121	0.131	103.77
2003	249,763	85.05	733	42,815	13.925	0.117	100.91
2004	254,643	86.42	722	43,739	15.117	0.129	97.85
2005	261,571	88.42	713	43,090	15.165	0.128	91.61
2006	267,769	88.95	781	45,956	14.435	0.125	92.42
2007	273,336	90.55	797	45,795	14.702	0.127	95.74
2008	272,742	93.74	832	46,105	14.562	0.130	92.18
2009	265,750	92.92	839	45,837	15.432	0.136	98
2010	273,647	93.82	862	46,168	15.571	0.138	97.57
2011	275,984	95.15	859	45,598	16.147	0.141	95.45
2012	285,939	96.99	888	46,801	16.289	0.141	94.46
2013	293,523	97.76	993	49,156	15.025	0.134	96.3
2014	303,762	98.93	1,011	49,410	15.236	0.137	100.6
2015	311,256	98.63	1,042	50,381	15.136	0.140	100.89
2016	323,490	100	1,093	52,392	15.359	0.141	100
2017	331,903	100.62	n.a.	n.a.	n.a.	n.a.	95.24
Unit	NT$/capita	%	NT$/case	NT$/case			%

Remarks: n.a. denotes not available.

**Table 4 ijerph-16-01206-t004:** The correlation coefficient between the medical capacity provided and the relevant impact factors.

Medical Capacity per Capita	Disposable Income per Capita *y*	Price Index *p*	Expense per Case *m*		Occurrence Rate *q*		Total Costs *h*
			Outpat.	Inpat.	Outpat.	Inpat.	
beds	0.8434 **	0.9142 **	0.8023 **	0.8493 **	0.7734 **	0.8009 **	–0.4578 **
workforce	0.9534 **	0.9789 **	0.9611 **	0.9131 **	0.6390 *	0.8081 **	–0.3392 **
physicians	0.9619 **	0.9805 **	0.9725 **	0.9111 **	0.6060 *	0.7987 **	–0.3170 **
nurses	0.9523 **	0.9795 **	0.9561 **	0.9202 **	0.6365 *	0.8024 **	–0.3450 **

Remarks: The superscript ** represents a 99% significance level, and * for a 95% significance level.

**Table 5 ijerph-16-01206-t005:** The occupancy rate of beds in 2017 (unit: %).

General Beds		Special Beds			
Acute General Beds	68.07	Intensive Care beds	72.55	Subacute Respiratory Care Beds	61.30
Psychiatric Acute General Beds	82.37	Burn Care Beds	41.36	Psychiatric Intensive Care Beds	94.51
Chronic General Beds	54.48	Burn Intensive Care Beds	54.63	General Isolation Beds	59.06
Psychiatric Chronic General Beds	91.58	Infant Care Beds	55.10	Positive Pressure Isolation Beds	75.19
Chronic T.B. Beds	20.14	Nursery Beds	43.06	Negative Pressure Isolation Beds	49.05
Leprosy Beds	74.93	Palliative Care Beds	60.94	Bone Marrow Transplantation Beds	49.72
International Health Care Beds	0.37	Chronic Respiratory Care Beds	79.68	Post-Acute Care Beds	23.46
				Integrated Medicine Post Emergency Department Beds	74.56

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
