# Peer review of "An Examination of the Determination of Medical Capacity under a National Health Insurance Program"

_ijerph, 2019, doi:10.3390/ijerph16071206_

Round 1

Reviewer 1 Report

49-51 If you place the footnote number, you don’t need to list the authors.

52- Throughout- It appears that APA and Journal methods are mixed.

87-88- This sentence needs a citation.

91- Place a Table with the symbols and definitions here.

184- What is the relevance of this Figure. Also, is lumping other providers with nurses appropriate.

With more procedures being done on an out-patient basis, wouldn’t the number of beds naturally decrease?

292- How did these increases compare with the GNP.

307-310- As mentioned earlier, the changing nature of reliance on outpatient procedures and treatments could have affected this.

Author Response

My Responses to Reviewer 1:

At first, we appreciate very much for the reviewers’ comment that is valuable and helpful for me to improve this article. In order to improve the quality of English writing, we have asked the editor office to edit it again and hope that the revised version of this article can meet the standard of this journal.

Comment: 49-51 If you place the footnote number, you don’t need to list the authors.

Response: Thanks for your comments. The revision was made accordingly.

Comment: 52- Throughout- It appears that APA and Journal methods are mixed.

Response: Thanks for your comments. The revision was made accordingly.

Comment: 87-88- This sentence needs a citation.

Response: The new reference has been cited on Line 86.

Cobb, C.W.; Douglas, P.H. A theory of production. The American Economic Review. 1928, 18, 139–165.

Comment: 91- Place a Table with the symbols and definitions here.

Response: A table to describe the symbols is added on P. 4 and a new sentence is inserted, reading as “The definition of symbols is described in Table 1.”

Table 1. The list of symbols

Symbols

Definition

A parameter between   the consumption of private and medical services

Private goods

Substitution parameter

g

The provision of   medical capacity

y

Disposable income

The consumer price   index

The proportion of

The disease occurrence   rate

Copayment paid by   patient

The amount of people   covered by insurance plan

The revenue of medical   service

The medical   expenditure for each treatment paid by the National Health Insurance

Comment: 184- What is the relevance of this Figure. Also, is lumping other providers with nurses appropriate.

Response: Thank you for your comment. In order to explain Figure 1 more detailed, several sentences are appended to the end of Line 186, and reads as

The high growth rate of nurses compared to physicians and beds may attribute to the expanding scopes of practice for nurses (Hall, et al., 2006; McKenna, et al., 2019). The rapid development of medical technology also increase the nurse’s working load. Considering the increase in the utilization efficiency of human resources, hospitals prefer the scopes of practice to be overlapping and flexible (White et al., 2009). Some administrative tasks have been shifted to nurses to reduce the health care costs (Fairman et al., 2011). Thus, the growth rate of nurses is much higher than that of physicians.

Hall, L.M.; Pink, L.; Lalonde, M.; Murphy, G.T.; O'Brien-Pallas, L.; Laschinger, H.K.S.; Tourangeau, A.; Besner, J.; White, D.; Tregunno, D.; Thomson, D.; Peterson, J.; Seto, L.; Akeroyd, J. (2006). Decision making for nurse staffing: Canadian perspectives. Policy, Politics, and Nursing Practice. 2006, 7, 261-269.

White, D.; Jackson, K.; Besner, J.; Suter, E.; Doran, D.; McGillis Hall, L.; Parent, K. (2009). Enhancing nursing role effectiveness through job redesign. Calgary, AB: Alberta Health Services.

Fairman, J.A.; Rowe, J.W.; Hassmiller, S.; Shalala, D.E. Broadening the scope of nursing practice. New England Journal of Medicine. 2011, 364, 193-196.

McKenna, L.; Wood, P.; Williams, A.; O’Connor, M.; Moss, C.; Griffiths, D.; Phillip, D.; Endacott, R.; Cross, W. Scope of practice and workforce issues confronting Australian Enrolled Nurses: A qualitative analysis. Collegian. 2019, 26, 80-85.

Comment: With more procedures being done on an out-patient basis, wouldn’t the number of beds naturally decrease?

Response:

In practice, the economic growth may stimulate the public to be more concerned on health status and as a consequence more outpatient cares and body examination are demanded. The preventive activities adopted by the public may detect the diseases at earlier days and reduce the length of stays in hospitals. Eventually, the number of beds may decrease. However, the occurrence of diseases depends many factors including dieting, sports, living styles, mental status etc. And thus, it cannot safeguard the decrease in the provision of beds.

Comment: 292- How did these increases compare with the GNP.

Response: I have updated the figure to describe the relationship between per capita medical expenditures and disposable income. Two sentences are inserted on Line 309 to illustrate Figure 2. and read as

A positive relationship between per capita medical expenditures and disposable income is found. This implies that the economic development may push up the demand for medical services.

Figure 2. The trend of per capita medical expenditure and disposable income 

Comment: 307-310- As mentioned earlier, the changing nature of reliance on outpatient procedures and treatments could have affected this.

Response: Thanks for your comment. I agree your point so a sentence is inserted on Line 331 to describe the changing of occupancy rates and reads as

In practice, the economic growth may stimulate the public to be more concerned on health status and as a consequence more outpatient cares and body examination are demanded. The outpatient cares may detect the existence of major diseases at earlier days and avoid the demand for beds. However, the aging society is characterized with the high demand for the treatment of chronic diseases and push up the demand for beds”.

Reviewer 2 Report

The paper makes a well-founded study exploring a theoretical model for the determination of the optimal medical capacity. For these purposes the article describes the results of a sensitivity analysis using the empirical data provided by the government of Taiwan.

In this regard, further information on the government data as shown in Table 2 is missed. Furthermore, some minimal assessment of the quality of government data and the absence of any bias would  be more than welcome. From a methodologically point of view, a limitation section or paragraph should be explicitly included in the paper.

The research selected scope is Taiwan. In order to avoid any confusion, it is recommended to replace the term "in a given region" in the abstract  (line 10) with "at a national level", as it appears in line 65.

We recommend to include in section 3 (research methods) some additional reference to the total population of Taiwan as well as to the insured population. In this way, the reader can get a more accurate idea of the selected scope.

Any case, this is an interesting paper. It is is really helpful for further developments concerning other countries. As the author highlights, future research might be extended to an analysis on the interface between the impact of demographic features and the demand for medical services.

Author Response

My Responses to Reviewer 2:

At first, we appreciate very much for the reviewers’ comment that is valuable and helpful for me to improve this article. In order to improve the quality of English writing, we have asked the editor office to edit it again and hope that the revised version of this article can meet the standard of this journal.

Comment: In this regard, further information on the government data as shown in Table 2 is missed. Furthermore, some minimal assessment of the quality of government data and the absence of any bias would be more than welcome. From a methodologically point of view, a limitation section or paragraph should be explicitly included in the paper.

Response: A paragraph is appended to the end of Section 3 to describe the limitation of the data provided by the government, and reads as

In fact, the relevant data released by the government does not fully reveal the limitation of data calculation. For example, the occurrence rate of diseases is calculated based on the visit per beneficiary issued by NHI. The cases to home nursing care, psychiatric rehabilitation, medical examination referrals commissioned by medical institutions, and repeated prescriptions for patients with chronic illnesses are excluded in the occurrence rate of outpatient cares. The inappropriate survey or incorrect collection technique may lead to bias. Fortunately, the data involving the medical expenditure, the occurrence rate of inpatient and outpatient, and the total costs is collected by the government through the insurance program system by the census. It means that the sampling error can be avoided. Furthermore, the insured rate has reached to 99.6%, implying the data collected may fully reflect the reality.

Comment: The research selected scope is Taiwan. In order to avoid any confusion, it is recommended to replace the term "in a given region" in the abstract (line 10) with "at a national level", as it appears in line 65.

Response: Thank you for your comment. The revision was made accordingly.

Comment: We recommend to include in section 3 (research methods) some additional reference to the total population of Taiwan as well as to the insured population. In this way, the reader can get a more accurate idea of the selected scope.

Response: I had added a sentence to describe the total population of Taiwan as well as to the insured population on Line 150. It reads as

Until now, the amount of population are 23,978,460 and the amount of population registered by NHI are 23,832,550 [31]. This implies that insured rate reached to 99.6%. 

Comment: Any case, this is an interesting paper. It is really helpful for further developments concerning other countries. As the author highlights, future research might be extended to an analysis on the interface between the impact of demographic features and the demand for medical services

Response: Thank you for your comment.

This manuscript is a resubmission of an earlier submission. The following is a list of the peer review reports and author responses from that submission.

Round 1

Reviewer 1 Report

I do not know the requirement of this journal, but I think that a complete section devoted to the state of the art would be convenient necessary. This would give a little more detail on the methodology and results of the literature.

The author makes the hypothesis of "representative citizen". It is known that this hypothesis does not work very well in practice. I know it's hard to work without that assumption. Consequently, I won’t suggest doing the work again without it. However, the author can discuss a little about the robustness of his/her results with respect to this hypothesis.

The utility function used by the author is a special one: the authors should precise its name and give a reference. In addition, why using this one and not another? I do not say to test all the possible utility functions, but doing research is not only test one thing, but it is having a global view of the problem (theoretically and empirically). Thus, the author can use and test only one function, but he/she should explain firstly why this one would be the most appropriate, and secondly discuss a bit about what happens if another one was used (robustness).

Initially, the work is very theoretical. However, the author was able to apply on real data leading to interesting results.

In such aggregate data, there is no standard error. However, the author carries out a sensitivity analysis that allows us to check the sensitivity of the results to a change in the input of the model. That is fine.

An important lack is that the author does not compare its method to other methods of the literature to check if it works better or not; or if it provides additional/different conclusions or not. However, since the state of the art is not very detailed, it is difficult to see what can be done, and perhaps my remark does not make sense. But it would be fine if the author can reply to this point.

To conclude, I think the work is interesting, but it is too much self-focusing. The author must take a step back and locates hi/her work in the state of the art: what happens if the representative citizen does not hold? What happens if another utility function is used? Does his/her method work better or give other results than the other methods of the literature? …

Author Response

My Responses to Reviewer 1:

At first, we appreciate very much for the reviewers’ comment that is valuable and helpful for me to improve this article. In order to improve the quality of English writing, we have asked the editor office to edit it again and hope that the revised version of this article can meet the standard of this journal.

Comment: The author makes the hypothesis of "representative citizen". It is known that this hypothesis does not work very well in practice. I know it's hard to work without that assumption. Consequently, I won’t suggest doing the work again without it. However, the author can discuss a little about the robustness of his/her results with respect to this hypothesis.

Response: Thank you very much for your valuable comments. A new paragraph was added and inserted into Section 5. Discussion. The new paragraph starts from Line 239 in the old version of this article, and reads as:

This paper considers the citizen with mean values of the demographic characteristics to represent the society. This implies that the distribution of the demographic characteristics is normal. If the gap between mean and median value increases, the optimal capacity needs to be adjusted. For example, the mean and median value of Taiwan’s personal disposable income in 2016 are NT$ 323,490 and NT$ 310,807, respectively. Compared the optimal capacity derived by median value and mean value, the following equation derived from Eq. (10) is obtained. 

               (13)

where the  and  are the mean and median disposable income, and  and  are optimal capacity obtained by the median and the mean value, respectively. Applying the data of disposable income, disease occurrence rates and medical expenditure appeared in Table 2 to Eq. (13) yields that  96.36%. Such a result implies that the optimal capacity may be reduced by 3.67% if calculated by median values compared to mean values. Basically, the high socioeconomic inequalities may provide negative impact on the maintenance of a health care system. As the gap between the mean and the disposable income is only 3.9%, the economic inequality seems not so serious in Taiwan. And thus, the optimal value of medical capacity based on a mean value may be acceptable.

Comment: The utility function used by the author is a special one: the authors should precise its name and give a reference. In addition, why using this one and not another? I do not say to test all the possible utility functions, but doing research is not only test one thing, but it is having a global view of the problem (theoretically and empirically). Thus, the author can use and test only one function, but he/she should explain firstly why this one would be the most appropriate, and secondly discuss a bit about what happens if another one was used (robustness).

Response: Thank you for your suggestion. Following sentence was inserted into the first paragraph of Section 2, starting on Line 66 in the old version of this article.

The social welfare is measured by a utility function that defines consumer’s preferences for goods or services. A utility function is generally classified into Perfect Substitutes, Perfect Complements, and Cobb-Douglas utility function. Solow (1956) presented the constant elasticity of substitution (CES) function for the application in economics fields to model production functions or ordinary consumer choice problems. Later Arrow et al. (1961) applies the CES utility function in consumer theories to describe the preference characterized by a constant elasticity of substitution between two differentiated goods or services. The CES function owns the relative advantage as it covers a much broader spectrum of substitutability between x and y. For ρ = -1, the function collapse to linear functions, representing perfect substitutes. For ρ → ∞, the curve of CES functions is L-shaped, representing perfect complements. When ρ approaches 0, the CES function converges to the Cobb– Douglas function. As the CES utility function has been widely accepted in research fields, this paper adopted it to serve for the measurement of social welfare. And thus, a constant elasticity of substitution between the provision of medical capacity g and the consumption of private goods Q is employed.

Reference:

Solow, R. M. (1956). A contribution to the theory of economic growth. The Quarterly Journal of Economics70(1), 65-94.

Arrow, K. J., Chenery, H. B., Minhas, B. S., & Solow, R. M. (1961). Capital-labor substitution and economic efficiency. The Review of Economics and Statistics, 43(3), 225-250.

Comment: An important lack is that the author does not compare its method to other methods of the literature to check if it works better or not; or if it provides additional/different conclusions or not. However, since the state of the art is not very detailed, it is difficult to see what can be done, and perhaps my remark does not make sense. But it would be fine if the author can reply to this point.

Response: Thank you for your suggestions. In fact, I have accessed to Science Direct (SDOL) and searched the relevant literature by means of the keywords including “determination of medical capacity”, “medical capacity planning”, and “capacity of medical services”. The outcome of the searching shows very little literature focuses on the determination of medical capacity as what we do. 

Several studies examine whether the provision of medical care is sufficient to meet the demand by the method of surveying directly to hospitals (e.g. Allorto et al., 2018; Touray et al., 2018). But, these paper just report the data of medical capacity.  For example, Touray et al, (2018) visit eight public health facilities providing secondary and tertiary care in Gambia. A questionnaire including the presence of an intensive care unit (ICU), the number of critical care beds, monitoring equipment, and the ability to provide basic critical care services is presented to the designated respondent. Their result concludes that the capacity of ICU is low and human resources and equipment are lacking. Some researches design a questionnaire to consult with experts by using Delphi method. For example, Alberti et al. (2018) aims to optimize the prescription center capacity by conducting a Delphi method that aiming at increasing access to therapy for patients with chronic hepatitis C virus (HCV) infection.

Some previous studies employ mathematical models to determine the location of medical facilities. I have mentioned in the original version of this article. Please see the third paragraph in “Section 1. Introduction”, starting from Line 40. 

Some researchers analyze the response of hospitals to the environmental change like fixed-fee payment systems and find the hospital may shift to more variable cost structure based on the financial perspective (Kallapur and Eldenburg, 2005; Holzhacker et al., 2015).

Although some studies aim at the solution for the capacity planning problem, they completely focus on a micro-level (e.g. Sariyer, 2018; Gai et al. 2018; Malik and Khan, 2015; Jiang and Seidmann, 2014; Yi, et al., 2010; Balakrishnan et al., 2007; Eldenburg et al., 2011).  For example, Sariyer (2018) employs the newsvendor model in planning the capacities of emergency call centers. In order to reduce the prevalence of preventable childhood diseases in Nigeria, Gai et al. (2018) present a mathematical model to determine hub locations by optimizing a shortest path and to obtain the optimal capacity of vaccine at the hubs.

Considering the methods adopted by the previous studies are almost different from this article, I appended a new paragraph to the first paragraph in “Section 6. Concluisons”. The new paragraph, starting from Line 305 (the old version) reads as:

Although some studies attempt to solve the solution for the capacity planning problem, they completely focus on a micro-level (e.g. Sariyer, 2018; Gai et al. 2018; Malik and Khan, 2015; Jiang and Seidmann, 2014; Yi, et al., 2010).  This paper attempts to determine the optimal medical capacity on a state level rather than a hospital level based on a systematic perspective. The model presented in this paper integrates the perspectives of the policy maker, the supplier (medical service providers), the demanders (the patients), and the health-care insurance systems. And thus, it provides more valuable information for policy makers to make adjustment on the medical care policy.  

Reference

Allorto, N. L., Wall, S., & Clarke, D. L. (2018). Quantifying capacity for burn care in South Africa. Burns Open2(4), 188-192.

Balakrishnan, R., Soderstrom, N. S., & West, T. D. (2007). Spending patterns with lapsing budgets: Evidence from U.S. army hospitals. Journal of Management Accounting Research19(1), 1-23.

Eldenburg, L. G., Gunny, K. A., Hee, K. W., & Soderstrom, N. (2011). Earnings management using real activities: Evidence from nonprofit hospitals. The Accounting Review86(5), 1605-1630.

Gai, D. H. B., Graybill, Z., Voevodsky, P., & Shittu, E. (2018). Evaluating scenarios of locations and capacities for vaccine storage in Nigeria. Vaccine36(24), 3505-3512.

Holzhacker, M., Krishnan, R., & Mahlendorf, M. D. (2015). The impact of changes in regulation on cost behavior. Contemporary Accounting Research32(2), 534-566.

Jiang, Y., & Seidmann, A. (2014). Capacity planning and performance contracting for service facilities. Decision Support Systems58, 31-42.

Kallapur, S., & Eldenburg, L. (2005). Uncertainty, real options, and cost behavior: Evidence from Washington state hospitals. Journal of Accounting Research43(5), 735-752.

Malik, M. M., Khan, M., & Abdallah, S. (2015). Aggregate capacity planning for elective surgeries: A bi-objective optimization approach to balance patients waiting with healthcare costs. Operations Research for Health Care7, 3-13.

Sariyer, G. (2018). Sizing capacity levels in emergency medical services dispatch centers: Using the newsvendor approach. The American Journal of Emergency Medicine36(5), 804-815.

Touray, S., Sanyang, B., Zandrow, G., Dibba, F., Fadera, K., Kanteh, E., ... Sanyang, A. (2018). An assessment of critical care capacity in the Gambia. Journal of Critical Care47, 245-253.

Yi, P., George, S. K., Paul, J. A., & Lin, L. (2010). Hospital capacity planning for disaster emergency management. Socio-Economic Planning Sciences, 44(3), 151-160.

Comment: To conclude, I think the work is interesting, but it is too much self-focusing. The author must take a step back and locates hi/her work in the state of the art: what happens if the representative citizen does not hold? What happens if another utility function is used? Does his/her method work better or give other results than the other methods of the literature?

Response: Thank you very much for your comment. In “Section 1” the paragraph starting from “The impact factors affecting the optimal capacity for the provision …” (Line 52) was revised, reading as:

The impact factors affecting the optimal capacity for the provision of medical services may provide valuable information for policymakers. However, many scholars focus on the micro-level to analyze the medical capacity planning problem. For example, Sariyer (2018) employs the newsvendor model in planning the capacities of emergency call centers. Jiang and Seidmann (2014) focus on the design of efficiency managerial contracts and capacity planning for medical facilities and analyze the impact of the linear contracting structure on capacity decisions, service levels, service volumes, and the allocations of costs. Yi et al. (2010) develop a generic simulation model to obtain steady-state hospital capacities. In order to reduce the prevalence of preventable childhood diseases in Nigeria, Gai et al. (2018) present a mathematical model to determine hub locations by optimizing a shortest path and to obtain the optimal capacity of vaccine at the hubs. Some researches design a questionnaire to consult with experts by using Delphi method. For example, Alberti et al. (2018) aims to optimize the prescription center capacity by conducting a Delphi method that aiming at increasing access to therapy for patients with chronic hepatitis C virus (HCV) infection.

This paper attempts to determine the optimal medical capacity based on a national level through a systematic perspective. A mathematical model is presented to investigate the impact factors by integrating the demand, the supply and the insurance program. The problem is formulated by …

Reference:

Alberti, A., Angarano, G., Colombo, M., Craxì, A., Di Marco, V., Di Perri, G., ... Pasqualetti, P. (2018). Optimizing patient referral and center capacity in the management of chronic hepatitis C: Lessons from the Italian experience. Clinics and Research in Hepatology and Gastroenterology. XXX, 1-11. https://doi.org/10.1016/j.clinre.2018.09.007

Gai, D. H. B., Graybill, Z., Voevodsky, P., & Shittu, E. (2018). Evaluating scenarios of locations and capacities for vaccine storage in Nigeria. Vaccine36(24), 3505-3512.

Jiang, Y., & Seidmann, A. (2014). Capacity planning and performance contracting for service facilities. Decision Support Systems58, 31-42.

Sariyer, G. (2018). Sizing capacity levels in emergency medical services dispatch centers: Using the newsvendor approach. The American Journal of Emergency Medicine36(5), 804-815.

Yi, P., George, S. K., Paul, J. A., & Lin, L. (2010). Hospital capacity planning for disaster emergency management. Socio-Economic Planning Sciences44(3), 151-160.

Reviewer 2 Report

Review of An examination of the determination of medical capacity under a national health insurance program

Thank you for the chance to review this piece. As a modeler myself for over 20 years, I am always very interested in the work of my peers. I have recommended significant modifications. I hope you will agree with my explanations. I think you would – in my position – recommend the same thing.

In your paper, you have set up the usual utility function with two constraints – the health spenders (ie consumers) and health system. You turn the crank, and get some first order conditions. So far, so good (even though I guess you will agree with me the model is nothing to write home about – we have all been there).

Next, you COMPARE the way these variables affect your g and some PAIRWISE correlations with Taiwan’s data.  I assume these are standard Pearson correlation coefficients without any bells or whistles.

IF I’VE UNDERSTAND THE PAPER CORRECTLY (and I put it in bold as a caveat)

You’ve got some theoretical and methodological issues to deal with.

Theoretical

1. So what? These findings basically show the way we expect your variables to react in a constrained economy. No one contents that Taiwan does not face the constraints driving your model (and thus g). The signs on Table 1 pop-out automatically as a function of the math. But so what?

2. Externalities. You assume that the optimal g is what people think it should be. But we know that’s not true. Government spends MORE on health than what people want – because of externalities. Because the elasticity parameter is the same across all goods, you can’t fudge your way out of this problem and say ‘well, its captured in the preference parameter – and indeed, the preference parameter is just waved away (so why have it there at all, other than because academic economists love it).

3. Where is supply and demand? Your figure 1 illustrates demand for medical goods in Taiwan (ignoring all the problems with trying to tie these noisy data with what houses actually want as health care). Your Table 2 shows your supply side. But your variable “g” is not govered by supply and demand as any economist would understand it. There is no real p and Q on the illustrations we show to first year students. I get that resources get cleared via costs.

Empirical

1. You model medical capacity (basically health care provision) has a system. Yet, the empirical methods you choose are simply bivariate correlation analysis. We have at least 40 system methods for dealing with this (from simple linear regression all the way to AMOS methods). I am 100% sure that you get such high correlations because you don’t use system methods. Once you change it and remove the ‘not-so-related parts’ from these correlations (like all the stuff they correlate with together), you will get more realistic coefficients of 0.04.

2. Correlations. If we use something as rudimentary as correlations, how do we know they are even statistically significant? You know, you could be looking at a rho of .9 and still its statistically insignificant.

Discussion.

From these obviously wrong correlations, we have discussion and recommendations for Taiwan’s medical system. Many of the assertions about how stingy the budget is, have not been explained. And the authors know their observations are completely speculative – by the hedging language used. I know its popular to put policy recommendations in economics papers. But if the recommendations are based on a correlation, and that’s it, its best to leave it out.

The thing about bed occupancy comes from no where. Either include it in the model and “correlation” analysis, or leave it out.

Language.

I know its cheap to talk about language. But it might be worth-while to have a GOOD native speaker go over the piece. The English is technically correct. But the phrasing is so tortured to a native speaker, that one really struggles. Just imagine if your paper is with 30 others. Friendly readability becomes important.

I am stopping here because its already a lot. There are other obvious things, like Figure 1 has 4-5 lines that all basically say the same thing. But I won’t go anymore into it – to see if these are issues can be tackled.

I am sorry I can not be more positive. I really have a soft spot for fellow modelers. But I guess no one cared enough to give useful feedback on much of your work in the past (I know, I work in Hong  Kong – and our Asian colleagues are not very confrontational).

Author Response

My Responses to Reviewer 2:

At first, we appreciate very much for the reviewers’ comment that is valuable and helpful for me to improve this article. In order to improve the quality of English writing, we have asked the editor office to edit it again and hope that the revised version of this article can meet the standard of this journal.

Comment: So what? These findings basically show the way we expect your variables to react in a constrained economy. No one contents that Taiwan does not face the constraints driving your model (and thus g). The signs on Table 1 pop-out automatically as a function of the math. But so what?

Response: I think that a mathematical model should be as simple as possible as the purpose of a mathematical model is just like a conceptual framework. The development of a model considers not only the correct reflection of the actual phenomenon, but also the simplification of the actual phenomenon. 

The mathematical model presented in this paper aims to describe the relationship between the optimal medical capacity planned by the policy maker and the relevant independent variables. This paper adopts the CES utility function that is widely accepted by many researches and applied in various fields. In fact, through a systematic consideration, this paper integrates the perspectives of the policy maker, the supplier (medical service providers), the demanders (the patients), and the health-care insurance systems. And thus, the signs on Table 1 cannot be pop-out automatically without the model formulation and sensitivity analysis. The result of the sensitivity analysis can provide some valuable information for the policy maker. For example, the policy maker can judge how to adjust the medical expenditure m and insurance premium  based on Table 1.  

Comment: Externalities. You assume that the optimal g is what people think it should be. But we know that’s not true. Government spends MORE on health than what people want – because of externalities. Because the elasticity parameter is the same across all goods, you can’t fudge your way out of this problem and say ‘well, its captured in the preference parameter – and indeed, the preference parameter is just waved away (so why have it there at all, other than because academic economists love it).

Response: In this paper, the externality of disease or medical services is excluded as the externality is very difficult to measure. However, I have to clarify that the optimal medical capacity g is determined based on the maximization of social welfare by the policy maker. And thus, the price and quantity is not determined by the market as the price (copayment and medical expenditure) is determined by the government. This implies that the model presented in this paper have considered the externality if it exists.

Comment: Where is supply and demand? Your figure 1 illustrates demand for medical goods in Taiwan (ignoring all the problems with trying to tie these noisy data with what houses actually want as health care). Your Table 2 shows your supply side. But your variable “g” is not govered by supply and demand as any economist would understand it. There is no real p and Q on the illustrations we show to first year students. I get that resources get cleared via costs.

Response: The demand for medical service can be derived by maximizing Eq. (1) subject to the constraint (2). For the demander (the patient), the copayment c is the price for each medical service and the quantity is represented by the disease occurrence rate q. Each visit to medical service providers, the patient has to pay copayment c including the registration fee and the extra cost when the patient prefers higher quality of medicine or material to the standard one that is prescribed by the insurance program. In general, the copayment is very low for outpatient cares in Taiwan and the government complains very much about the overuse of medical services.

For the supplier, the sum m + c of medical expenditure m received from the insurance program and the copayment received from the patient is equivalent to the medical price. And, the disease occurrence rate q is equivalent to quantity. The market is monopolized by the insurance program reflecting the same price of medical services (medical expenditure paid by the insurance program and the copayment paid by the patient) across service providers. The cost for each item of medical service should be reported by all the service providers and analyzed by the insurance program. The standard price for each item of medical service is negotiated by the service providers and the insurance program, and eventually determined. For interested readers, the information about NHI fee (the standard price for each item of medical services) schedule released by MOHW (https://www.nhi.gov.tw/Content_List.aspx?n=58ED9C8D8417D00B&topn=D39E2B72B0BDFA15) can be obtained.

And thus, this paper use Eq. (3) to represent the supply function for medical services. And thus, the data of expenditure m and occurrence rate q in Table 2 is a portion of the price for the supplier (the service provider) and quantity for medical services in this paper.

Comment: You model medical capacity (basically health care provision) has a system. Yet, the empirical methods you choose are simply bivariate correlation analysis. We have at least 40 system methods for dealing with this (from simple linear regression all the way to AMOS methods). I am 100% sure that you get such high correlations because you don’t use system methods. Once you change it and remove the ‘not-so-related parts’ from these correlations (like all the stuff they correlate with together), you will get more realistic coefficients of 0.04.

Response: I have explained in the response to your comment “Theoretical 1” that the model is formulated through a systematic perspective based on Taiwan’s medical service system. I believe that the variables appeared in the model should be related theoretically.

Comment: Correlations. If we use something as rudimentary as correlations, how do we know they are even statistically significant? You know, you could be looking at a rho of .9 and still its statistically insignificant.

Response: Sorry to make you confused. The significance level is added into Table 3. And the revised Table 3 is listed below for your reference.

Table 3. The correlation coefficient between the medical capacity provided and the relevant impact factors     

Medical

capacity per capita

disposable income per capita y

price index

P

expense per case m

occurrence rate q

Total costs h

Outpat.

Inpat.

Outpat.

Inpat.

beds

0.8434**

0.9142**

0.8023**

0.8493**

0.7734**

0.8009**

-0.4578**

workforce

0.9534**

0.9789**

0.9611**

0.9131**

0.6390*

0.8081**

-0.3392**

physicians

0.9619**

0.9805**

0.9725**

0.9111**

0.6060*

0.7987**

-0.3170**

nurses

0.9523**

0.9795**

0.9561**

0.9202**

0.6365*

0.8024**

-0.3450**

Remarks: The superscript ** represents for 99% significance level, and * for 95% significance level.

Comment: From these obviously wrong correlations, we have discussion and recommendations for Taiwan’s medical system. Many of the assertions about how stingy the budget is, have not been explained. And the authors know their observations are completely speculative – by the hedging language used. I know its popular to put policy recommendations in economics papers. But if the recommendations are based on a correlation, and that’s it, its best to leave it out.

The thing about bed occupancy comes from no where. Either include it in the model and “correlation” analysis, or leave it out.

Response: I agree that Table 3 in the original version of this article without indicating the significance level is not good. However, the result of the correlation test is not obviously wrong.

Due to my bad English, I cannot fully comprehend what you mean “many of the assertion about how stingy the budget is”. Could you please explain more for my revision. Many thanks.

Comment: Language.
I know its cheap to talk about language. But it might be worth-while to have a GOOD native speaker go over the piece. The English is technically correct. But the phrasing is so tortured to a native speaker, that one really struggles. Just imagine if your paper is with 30 others. Friendly readability becomes important.

Response: Really, language is a big problem to non-English speakers. I agree with you that readability is very important for readers. I also know my English level may be criticized by reviewers. And thus this paper has been edited by an English native speaker before submitting to IJERPH. Please see attached the invoice and the certificate, dated 2018/10/24. After reviewing the edited version of my article by American Journal Experts, I feel the readability of this article should be OK. Furthermore, IJERPH is an international journal and serves the interested readers in the world including non-English speaking countries. Too many native slangs may make English native speakers pleased, but bring about misunderstanding for non-English speakers. Of course, I believe you may be right in judging the level of readability. I sincerely request you to recommend a good native speaker who works professionally for English edition for my contact. Thank you very much. 

Comment: I am stopping here because its already a lot. There are other obvious things, like Figure 1 has 4-5 lines that all basically say the same thing. But I won’t go anymore into it – to see if these are issues can be tackled.

Response: Thank you very much for your de-bugging. This paragraph starting from “By the end of 2017, …” (Line 144 in the old version) was revised. The revised reads as

As physicians and nurses play the primary role in providing medical services in addition to hospital beds, this paper considers (1) the number of beds, (2) the total number of individuals in the workforce, (3) the number of physicians, and (4) the number of nurses representing the capacity provided in practice. By the end of 2017, medical care institutions that offer general beds, special beds, specially designated beds and beds in clinics provide a total of 164,590 beds. Figure 1 indicates the historical trend of the number of workforce employees, physicians, nurses, and beds per 10,000 people during 1998-2017.

In 2017, 10,000 people can be served by 69.83 beds, marking an increase from 56.80 beds in 1998 [21]. By the end of 2017, 251,041 individuals in the workforce were practicing in the health profession, including 46,311 physicians, 6,685 traditional Chinese medicine doctors, 14,379 dentists, 14,695 pharmacists, and 122,063 nurses and other professionals [25]. This means that the total workforce per 10,000 population increases from 55.41 in 1998, including 12.34 physicians and 16.31 nurses, to 106.50 in 2017, including 19.65 physicians and 51.78 nurses. Figure 1 demonstrates that the growth rate of beds is much lower than that of personnel. The growth rate for the provision of beds per capita from 1998-2017 is 22.94%, which ranks lowest. In contrast, the number of nurses increases 217% for the same period, which ranks highest.

Round 2

Reviewer 2 Report

Thanks for taking the time to consider my comments!

I am willing to overlook the math model and the other points you raise (the marketplace of ideas can decide). 

However, I can not overlook the lack of a system-based method of estimation (like structural equations modelling or at least vector regression). The correlations are just plain wrong if you do not adjust for the way that variables interact with each other.

I am sorry to speak so bluntly, but let me say it clearly to avoid misunderstanding...

* I do not need a letter of explanation and excuses, I need to see a manuscript which addresses the points I raise. Even if you feel my comments are wrong, they need to be addressed in the paper, not in private communication....*

I sound very harsh in writing; I want too friendly. But its the clearest way I can say it. If you revise, please do not send comments. Just revise the paper please :)

Like I said, I am happy to work with you, as long as the editors allow! I also do not like it when reviewers make troubles for the papers I submit to journals. I just hope that - after thinking about it and talking it over with modellers you admire - you will see how this will help you in the longer term!